# Measures for the Improvement of Feasibility Studies and Investment Reviews: Identification and Verification of Major Project Sectors Considering Balanced Regional Development

**Heecheol Shim [1] and Jaehwan Kim [2,*]**

[1]   Public Investment Analysis Center, Chungbuk Institute, Cheongju 25817, Chungbuk, Korea; sim@cri.re.kr
[2]   Department of Real Estate Studies, Kongju National University, Yesan, Gongju-si 32439, Chungnam, Korea
*   Correspondence: jaehwan@kongju.ac.kr; Tel.: +82-(0)41-330-1402

**Abstract:** This study identified the balanced development indicators that affected the results of the 2019 central investment review of local financial investment projects in South Korea. Factors with positive B values and categorized under the sectors of safety, health, and social welfare were given greater weight during the investment review. Based on the empirical analysis results and verification of the findings using sector-specific weights, this study proposed measures to improve investment reviews of local financial projects considering balanced regional development. We believe that our study makes a significant contribution to the literature because there is a lack of empirical studies on the topic, especially those using sector-specific weights based on investment review criteria. Further, we believe that this paper will be of interest to the readership of your journal because it addresses balanced regional development, which is considered a prerequisite for sustainable economic growth.

**Keywords:** feasibility studies; balanced regional development; real estate development; local financial investment project; evaluation criteria

## 1. Introduction

For local financial investment reviews, the Republic of Korea conducts feasibility studies as a pre-inspection tool to ensure the rational and efficient use of funds for public investment projects and the prevention of negligent duplicate investments [1]. Feasibility studies are largely divided into two types: preliminary ones in accordance with the National Finance Act and those in accordance with the Local Finance Act. The present study addresses the latter type. Feasibility studies are also divided into economic and policy analyses [2]. Economic analysis mainly involves a cost-benefit analysis, and policy analysis qualitatively describes social values and unquantifiable factors. However, to comprehensively determine the feasibility of public investment projects, these two analyses must be integrated, which is a difficult task. This is because the indicators used in each analysis are divided into metrics and non-metrics [3]. Thus, existing feasibility studies in accordance with the Local Finance Act have presented two separate sets of analysis results. To resolve the differences between these factors as much as possible, preliminary feasibility studies under the National Finance Act have been performed with the provision of additional points to underdeveloped areas considering balanced regional development, together with the analysis results of the benefit/cost (B/C) ratio and analytic hierarchy process (AHP) [4]. The results have accounted for the fact that B/C ratios are inevitably low in regions with relatively small populations and weak infrastructure.

Preliminary feasibility studies have continuously increased the degree of representation of balanced regional development. However, in April 2019, a reform measure was proposed to dualise and evaluate metropolitan and non-metropolitan areas in a differentiated manner [5]. In addition, establishing methods that represent balanced national development in preliminary feasibility studies was presented as a specific task; efforts to represent balanced national development continue to be made in the preliminary budget review process for national projects [6].

Feasibility studies in accordance with the Local Finance Act include the degree of regional development in policy analysis items, but they present only the results of individual indicators. This makes it difficult to argue that the government sufficiently examines balanced regional development. However, investment reviews present standards for a comprehensive assessment of the needs and urgency of the project, the level of fulfillment of residents' long-awaited benefits, and the level of project demand in accordance with the evaluation criteria [7]. Unlike in preliminary feasibility studies, the B/C ratio does not have an absolute impact on the investment review in feasibility studies in accordance with the Local Finance Act. Moreover, the Investment Review Committee makes the final decision, evaluating multiple factors in tandem. Therefore, it is necessary to assess whether actual investment reviews consider the degree of balanced regional development. If factors related to balanced regional development have been affecting investment reviews, then it can be considered that factors not represented in feasibility studies have been represented in investment reviews. Given that feasibility studies ultimately aim to support decision-making in investment reviews, it is necessary to identify and represent indicators relevant to regional balance manifested in the current investment review [8]. In addition, the figures should be presented objectively and quantitatively for the identified factors to ensure consistency and objectivity in the investment review. However, unlike preliminary feasibility studies, the scope of investment reviews of local financial projects is different from that of national projects. Moreover, there is a separate decision-making organisation called the Investment Review Committee, so it is reasonable to take a different approach. Furthermore, the assessment of local financial investment reviews must consider the sectors of investment projects, along with regional conditions. This is because unified evaluation criteria for all project areas do not sufficiently represent the nature of each project [9]. Therefore, this study has the following objectives. First, it seeks to examine analysis methods in South Korea and other countries for the application of methods for balanced regional development, in addition to reviewing their linkages with local financial investment projects. Second, based on the passage status of the 2019 central investment review of local financial investment projects, this study conducts an empirical analysis to determine which areas of the balanced development indicators of city, county, and district are represented. Third, based on research related to the application of sector-specific weights in local financial investment projects, this study verifies the results of the above analysis and presents ways to utilise them. Fourth, by integrating these results, this study aims to present areas in which balanced development indicators are being represented in local financial investment projects and to propose a methodology for improving the current evaluation system. This study seeks to determine the factors that have affected the central investment review in specific sectors and to identify the weight of each sector, verifying the weights through the analysis results. It aims to establish the reliability of investment reviews by presenting two sets of analysis results and verifying them with different analysis models.

## 2. Theoretical Examination

Existing studies related to investment reviews in South Korea have mainly focused on the purpose of improving systems. It is also necessary to pay careful consideration to the topic from an academic perspective. Most South Korean studies related to investment reviews have been conducted by scholars and government officials belonging to the Korea Development Institute, the Korea Research Institute for Local Administration, and local government-affiliated research institutions [10]. However, this study considers both academic and practical implications; it utilises a considerable amount of foreign research on the types and standardisation of investment projects for academic implications.

The previous research reviewed here can be divided into three main categories. The first category includes studies that have analysed the institutional problems of investment reviews in South Korea and proposed measures to improve the evaluation system, including that regarding balanced regional development. A study based on the Seoul Metropolitan Government's investment review case highlighted the need for an eventual linkage of investment review with other systems, including mid-term financial plans and budgets [11]. Project regions are areas from which internal and external economic ripple effects and additional job creation effects are derived, indicating close relevance to not only investment reviews but also other systems. Furthermore, examples of foreign investment reviews are introduced here, along with the presentation of non-metric factors as well as quantitative analysis as factors for evaluation. Because most studies have emphasised the B/C ratio for investment review projects, the present study attempts to present other aspects or measures to enhance the expertise of the investment review operation department and proposes new review techniques. Studies examining the soundness of local finance have argued for the rational allocation of various investment resources and the institutional supplementation and enhancement of independent investment reviews [12]. In addition, to improve the current investment review standards and to enhance the public nature and equity of investment assessment, such studies have proposed changes in point allocation methods by categorising projects subject to review as well as using an approach with different review and application methods. Studies identifying issues, and the rationale thereof, related to investment reviews have noted the lack of systematic review items, subjective and non-metric criteria, and rational evidence and arguments for each review item [13]. As methods for improvement, previous studies have argued for adjusting the scale of project costs, including projects that had been excluded from review due to hazards posed to mid- to long-term fiscal soundness (linkage to mid-term local financial plans), systematically improving review items, and ensuring balanced regional development. Previous studies have also examined the operational status of feasibility study system analysis and presented institutional issues and improvement plans for linkage with the investment review system [14]. Such improvement plans consider the reinforcement of local government capacity as a prerequisite—emphasising the indispensability of the review of local financial investment projects for the efficient implementation of projects—rather than as a means of control. As another improvement measure, some studies have redesigned and presented a checklist for investment reviews in terms of macro-national and balanced regional development [15]. In addition to the rationality and efficiency of the review criteria, previous studies have argued for the addition of review criteria based on the judgement that basic investment projects—namely, those related to the environment, welfare, safety, labour, and consumer protection—are important reference points in terms of social welfare [16]. They have also argued for the attainment of justice and reliability, and the enhancement of review and financial capacity of local governments by establishing a scientific and systematic model for review methods [16].

The second category includes studies classifying and standardising investment reviews according to the types of projects subject to review. These studies have argued that efficiency, equity, and environmental preservation, in addition to the classification of project types, should be considered for local investment projects, since they are local development projects that involve local government finances [17]. In other words, public development projects should demonstrate a certain level of efficiency. Furthermore, two categories of projects subject to investment review have been presented in such research [18]. Social functions are divided into daily, auxiliary, developmental, and social basic projects, indicating that projects should be divided in terms of social functions and evaluated with some degree of deviation by sector. Public interest and necessity are subdivided into essential public interest projects, essential private interest projects, public benefit projects, and private benefit projects. In other words, even among public projects, types of projects are classified by public and private interests and by purpose of pursuit. In addition, previous studies have argued that some profitability should be considered even in public projects to reduce the tax burden on local governments, under a classification of projects in terms of public nature and profitability [14]. A previous study

classified projects subject to review according to their social functions into, as mentioned earlier, daily, auxiliary, developmental, and social basic projects [19]. The study likewise analysed the prerequisite in terms of efficiency and equity [19]. In addition, public investment projects were divided into those regarding economic infrastructures, such as roads, electricity, water supply and sewerage, and bridges, and those regarding social infrastructures, such as childcare, environment, and sanitation. In other words, public investment projects can be largely divided into economic and social infrastructure from a macroscopic perspective of national development; the former is considered important in terms of efficiency, while the latter is important in terms of balance [20]. Previous studies have also categorised projects subject to review based on production and purchasing entities. In other words, the project categorisation is based on the pre-eminence of the producing entity in either the public or private interest, and the main forms of purchase of such services [21]. According to the nature of each urban infrastructure project, Roth's study [21] that classified projects into basic service projects (such as road and sewage treatment plants) and comfort service projects (such as parks and libraries) is considered to be in line with the context of the project classification presented in Hansen's study [20]. Another study classified investment review criteria by supply cost and scope of benefits. It presented the same context as the investment review currently in practice, in which the investment review entity changes based on factors such as the scale of the total project cost [22]. The study also argued that the impact area should be established and distinguished by the extent of the people benefiting from the facility. Like Smith [22], Barlow [23] also categorised public investment according to the magnitude of ripple effects. In addition, one study cited effectiveness, efficiency, appropriateness, responsiveness, equity, and adequacy as the main criteria for decisions on public investment project evaluation analyses [24]. It argued that equity and effectiveness should be especially considered for urban infrastructure classified as basic infrastructure facilities [24]. Another study proposed using weights for review methods largely based on four characteristics in terms of policy [25]. It presented a hierarchical structure chart of investment project sectors based on the results of previous studies, including the aforementioned ones and those related to the evaluation of local financial investment projects [25]. It has also been noted that the investment project model can be largely classified into an economic development model and a political stability model [26]. Therefore, from a comprehensive perspective, it can be classified into economic and social infrastructure [26]. In addition, one study presented efficiency, equity, and political practicability as important criteria based on the alternative evaluation of review criteria [27], while another study examined the categorisation of projects based on economic efficiency, social efficiency, and environmental feasibility [28]. These previous studies have argued that public investment projects must demonstrate some level of efficiency, without presenting clear distinctions between economic and financial efficiency. Nevertheless, they have also argued that efficiency cannot be completely ruled out because the government must examine operating expenses after the investment review to ease the financial burden on local governments. A previous study classified evaluation criteria into quantitative and qualitative criteria; it reported that quantitative standards include items such as efficiency, effectiveness, adequacy, and productivity, while qualitative criteria include items such as responsiveness, adequacy, and democracy [29]. It has also been argued that from a national economic perspective, local financial investment projects pursue the interests of the entire society and have significant ripple effects on the region [30]. Public investment must hence be made continuously in terms of ensuring adequacy and meeting rapidly changing social needs; therefore, it is possible to categorise the projects based on two criteria, namely, sustainability and adequacy of public investment projects [30]. Another previous study classified public investment projects into basic and optional investment projects, arguing that investment projects of intermediate characteristics can be classified in terms of urgency and obligation [10,31]. In sum, based on the previous research results, local investment projects are important in terms of equity and efficiency of the rational allocation of limited resources, and in terms of balance, equity, and sustainability of regional development.

The third category of previous research includes normative studies, which are centred on developing guidelines for the investment review process and targeting the development of objective and rational investment review techniques. To emphasise the method of weight calculation for the review criteria through project categorisation, previous studies have attempted to ensure differentiation in the evaluation system by weighting projects based on efficiency, equity, balance, and innovation [25]. To represent both efficiency and effectiveness, previous research has evaluated both efficiency and effectiveness of projects by presenting measures to enhance efficiency through economic analysis, which accounts for the largest percentage of investment review techniques [32]. The importance and validity of investment reviews have been considered for studying the review analysis device to ensure feasibility, while providing directions for improvement of the system through a detailed analysis of the review [33]. Furthermore, previous research has presented a standard model for calculating available investment resources to establish a model for the importance of local financial soundness and the smooth implementation of projects [33]. For the operation of preliminary feasibility systems, a previous study presented the following factors: guideline development, analysis items, analysis procedures, and measurement items [34]. Moreover, a study analysed the investment review process and identified the factors affecting the investment review through statistical analysis [35]. The results of that study empirically demonstrated that economic and financial evaluation, which had traditionally been emphasised, did not affect the decision to implement the project to the extent previously known. The study also reported that the most influential factors on the results of the investment review were as follows: financial issues, such as mid-term local financial plans related to funding; linkages among various regions; ripple effects related to the project; and opinions and compelling will of the investment review department. In addition, the types of projects, evaluation criteria, and working reports of the review and evaluation officers were selected as factors affecting the investment review decision. The results of the investment review (adequacy, conditionality, reassessment) were analysed as dependent variables, and the types of projects were divided into basic, intermediate, and selective investment projects depending on the nature of the project. The adequacy of project scale and cost was selected as evaluation criteria, and variables of internal decision-making, such as adequacy and conditionality, were selected as factors affecting the working report. The previous studies under this category have also delineated the importance of investment reviews in terms of national economic development and the ripple effects of local governments. They have also emphasised the need to avoid arbitrary and subjective decisions in investment reviews and to proceed on a more objective and reasonable basis.

Despite the numerous studies discussed above, there is relatively insufficient research on the introduction of balanced development indicators. In addition, regarding preliminary feasibility studies, there are concerns that reality is not considered, in spite of improvements to the system, such as continuous revisions and supplements. Therefore, this study attempts to identify which of the balanced development indicators in the current central investment review affect the review results. Subsequently, it integrates the results of a study by one of the present authors on the establishment of a sector-specific review system for the review of local financial investment projects [36]; this is with respect to unified evaluation methods and systems criticised in having failed to represent the characteristics of each sector. Following this, the present study verifies the above analysis results. There are existing weighted studies regarding national and regional development at a macro level, but studies with sector-specific weights based on investment review criteria are relatively scarce. Therefore, to overcome the limitations of the above-mentioned research, this study seeks to identify factors affecting the current balanced development indicators for city, county, and district units based on the passage status of projects subject to the 2019 central investment review. For verification, projects subject to the investment review were classified into 14 facilities based on the industry association table, which is the classification system used in previous research, and actual investment review project standards. In establishing the evaluation criteria, this study seeks to formulate a new evaluation system model by organising a hierarchical structure chart involving practical implications for actual investment reviews as well as academic

implications channelled through numerous studies. Lastly, to include practical implications, this study derives weights by sector based on supporting legal data currently being utilised as investment review criteria, in addition to establishing evaluation criteria by sector and verifying analysis results thereof.

## 3. Materials and Methods

Balanced development indicators were adopted from the data published by the National Balanced Development Information System (NABIS) in 2019. These were used to identify factors that were not quantified in actual guidelines or policy analysis but that affected the review [37]. In addition, in line with the intention to utilise data published in 2019, the analysis was conducted by categorising the passage status of each region (city, county, and district) based on 436 projects commissioned for the central investment review in 2019 [38]. Furthermore, to verify the analysis results, this study identified the weight of each sector in the investment review criteria. To this end, investment projects were classified into economic and social infrastructure; by primary and secondary priorities, in terms of the equity, effectiveness, balance, and efficiency of the infrastructure, a total of five groups were considered. Fourteen evaluation models were established for the subcategories of the investment project sectors: basic facilities; water supply and sewerage; environment and hygiene; roads and transportation; industries and small and medium enterprises; regional development; agriculture, forestry, and fisheries; health and medical care infrastructure; social welfare facilities; living environment (parks and green areas); fire and disaster prevention; culture and tourism; education and sports; and general administration. For the analysis method, the weights were calibrated by deriving the relative and absolute importance through AHP and fuzzy analysis. To compute subjective decision-making more objectively using the above methods, the integrated opinions of experts were quantified. To this end, a total of 72 survey questionnaires were distributed to experts by various methods, such as e-mail, face-to-face communication, and focus group interview (FGI), using purposive sampling. For the final analysis, the data included 63 effective responses from 70 recovered copies, excluding 7 that demonstrated statistical consistency of 0.1 or higher.

## 4. Analysis Results and Discussion

### 4.1. Results of the Empirical Analysis of the Central Investment Review

This study analysed 436 projects commissioned for the 2019 central investment review by a classification of passage status by city, county, and district (Table 1).

**Table 1.** Numbers and Passage Rates of Commissions for the 2019 Central Investment Review.

| Rounds | Number of Commissions | Passage Rate |
|---|---|---|
| 19–Irregular | 13 | 92.31% |
| 19–1st | 131 | 64.75% |
| 19–2nd | 106 | 76.42% |
| 19–3rd | 186 | 79.03% |
| Total | 436 | 78.13% |

Based on the even distribution of passage rates of local financial investment projects for each round, as presented in Table 1, the balanced development indicators established by NABIS were used to identify factors that were not represented in actual guidelines or economic analysis but that affected the review. The balanced development indicators, published by NABIS in 2019, consist of key indicators that enable the comparison of development levels between regions, and sector or objective indicators that demonstrate various living conditions. The key indicators include the 40-year average population growth rate (population indicator) and the 3-year average financial self-reliance (economic indicator); the sector indicators include those regarding housing, transportation, industry, jobs, culture, and life

satisfaction. These indicators are used for the future support of financial projects and deficiencies pursued by NABIS. They are major indicators that can be applied to policy evaluation and feedback. However, this study utilised balanced development indicators for city, county, and district units instead of all indicators. This is because balanced development indicators are not critically significant when analysed at the city and provincial levels. For example, even within Gyeonggi Province, Suwon City and Yeoncheon County demonstrate significant differences in regional development levels, and hence, all analyses were conducted at the city, county, and district levels. Therefore, among the NABIS balanced development indicators, the following indicators presented only at the city and provincial levels were excluded: the ratio of households below the minimum housing standards (%); the number of patents (counts); research and development (R&D) expenses per R&D personnel (1000 won/one R&D personnel); the number of audience seats in entertainment facilities per 1000 people (number of seats/1000 people); the number of art activities per 100,000 people (activity count/100,000 people); the number of residents per rescue worker (number of residents/one rescue worker); and the proportion of recipients of national basic livelihood security (%). Table 2, below, shows the selected key indicators, sectors of objective indicators, and names of project sectors and indicators.

**Table 2.** Classification of Selected Variables and Names of Indicators.

| Name of Variable | | Contents |
|---|---|---|
| **Dependent variable** | Passage status | Passage status of the central investment review; passage 1, non-passage 0 |
| **Independent variable subfactor** | | **Name of indicator** |
| Key indicators | Population | Average population growth rate (1975–2015) |
| | Economy | Three-year average financial self-reliance (2015–2017) |
| Objective indicators | Housing | Ratio of old housing |
| | | Ratio of vacant housing |
| | | Water supply rate |
| | | Sewerage supply rate |
| | Transportation | Road pavement rate |
| | | Highway interchange (ic) accessibility |
| | | High-speed rail accessibility |
| | | Percentage of population in parking lot service area (0.75 km) |
| | Industry and jobs | Rate of increase or decrease in the number of businesses in the last three years (2015–2017) |
| | | Rate of increase or decrease in the number of employees in the last three years (2015–2017) |
| | | Three-year average knowledge-based industry aggregation rate (2015–2017) |
| | | Percentage of regular workers |
| | Education | Number of childcare facilities per 1000 infants (0–5 years old) |
| | | Number of schools per 1000 school-age population (primary, middle, and high schools) |
| | | Percentage of infant population in day-care centre service area |
| | | Percentage of school-age population in elementary school service area |
| | Culture and leisure | Number of cultural infrastructure facilities per 100,000 people |
| | | Percentage of population in performing arts cultural facility service area |
| | | Percentage of population in library service area |
| | | Percentage of population in public sports facility service area |
| | Safety | Number of residents per 119 safety centres |
| | | Fire station accessibility |
| | | Police station accessibility |
| | Environment | Urban park area per 1000 people |
| | | Rate of green area |
| | | Air pollutant emissions per 1 $km^2$ |
| | | Percentage of population in residential park service area |
| | Health and welfare | Ratio of single-person households over 65 years of age |
| | | Expenses on social welfare and healthcare |
| | | Number of social welfare facilities per 100,000 people |
| | | Number of medical facility beds per 1000 people |
| | | Percentage of senior citizen population in senior citizen leisure and welfare facility service area |
| | | Percentage of population in emergency medical facility service area |
| | | Percentage of population in hospital service area |

According to a technical statistical analysis conducted prior to the full empirical analysis, 330 projects (75.7%) were found to have passed the investment review and 106 projects (24.3%) were found not to have passed. The logit model analysis method adopted the "enter" method. The analysis model demonstrated a −2 log-likelihood value of 81.41, while the Cox and Snell R-square and Nagelkerke R-square values explained 49.1% and 64.9%, respectively, of the variance (Table 3).

**Table 3.** Model Summary.

| Step | −2 Log-Likelihood | Cox and Snell R-Square | Nagelkerke R-Square |
|------|-------------------|------------------------|---------------------|
| 1 | 81.41 | 0.491 | 0.649 |

In addition, the goodness of fit of the model was determined using the Hosmer and Lemeshow test statistic values. The test demonstrated a Pearson chi-square statistic of 8.945 (Table 4). Moreover, the significance probability was 0.29, indicating a statistically significant goodness of fit of the model.

**Table 4.** Hosmer and Lemeshow Test.

| Step | Chi-Square | Degree of Freedom | Significance Probability |
|------|------------|-------------------|--------------------------|
| 1 | 8.945 | 8 | 0.029 |

This study sought to identify the balanced development indicators that affected the passage of projects in the central investment review. The variables with significant probabilities are summarised in Table 5.

**Table 5.** Analysis Results.

| Sector | Name of Indicator | B | S.E. | Wald | Significance Probability | Exp (β) | Comment |
|--------|-------------------|---|------|------|--------------------------|---------|---------|
| Transportation | High-speed rail accessibility | −0.21 | 0.006 | 12.098 | 0.001 | 0.979 | Downward indicator |
| Industry | Rate of increase or decrease in the number of businesses in the last three years (2015–2017) | −0.171 | 0.074 | 5.350 | 0.021 | 0.843 | Upward indicator |
| Education | Number of childcare facilities per 1000 infants (0–5 years old) | −0.083 | 0.040 | 4.382 | 0.036 | 0.920 | Upward indicator |
| Safety | Number of residents per 119 safety centres | 0.001 | 0.000 | 3.866 | 0.049 | 1.000 | Downward indicator |
| Safety | Fire station accessibility | −0.166 | 0.062 | 7.224 | 0.007 | 0.847 | Downward indicator |
| Safety | Police station accessibility | 0.372 | 0.131 | 8.086 | 0.004 | 1.450 | Downward indicator |
| Environment | Urban park area per 1000 people | 0.001 | 0.000 | 5.848 | 0.016 | 1.000 | Upward indicator |
| Environment | Percentage of population in residential park service area | −0.024 | 0.009 | 6.844 | 0.009 | 0.976 | Upward indicator |
| Health and welfare | Percentage of population in emergency medical facility service area | 0.013 | 0.006 | 4.091 | 0.043 | 1.013 | Upward indicator |

According to the analysis, the project passage rate was higher under the following conditions: greater high-speed rail accessibility; lower rate of increase or decrease in the number of businesses in the last three years; lower number of childcare facilities per 1000 infants (ages 0–5 years); greater number of residents per 119 safety centres; better fire station accessibility and lower police station accessibility; smaller urban park area per 1000 people; and greater percentage of population in emergency medical facility service areas. It is important to note the sectors that demonstrated high passage rates despite the improvement in balanced development indicators, as it can be considered

that the absolute necessity of the project is more important in those sectors. The project passage rate increases in the investment review as the high-speed rail accessibility improves because it was determined that the regional economic impact of various industries in the corresponding region will increase as the accessibility improves. Moreover, the passage rate in the investment review increases as the fire station accessibility increases because this is closely related to the actual safety of people. Fire station building projects are currently exempt from the investment review, and the safety of people with respect to fire or disaster can be interpreted as having an absolute value that cannot be determined by B/C ratio results or policy analysis results. Therefore, improved accessibility to fire stations indicates that minimal safety is guaranteed, and for investment projects, the passage rate of the investment review will increase if minimal safety facilities exist in the vicinity for emergency situations. Conversely, the higher passage rates for items with lower balanced regional development indicators possibly demonstrate the underdevelopment of the region or the local situation, which must be accounted for in future feasibility studies. The indicators presented in the analysis results and the commonalities in all areas, except for the aforementioned areas where the balanced regional development indicators increase, can be examined in two major policy trends: regional extinction inhibition and local economic vitalization aspects. First, the investment review aims to build basic infrastructure in order to inhibit regional extinction. Particularly, regional extinction caused by low birth-rate and aging population has emerged as a persistent social problem. Therefore, as an alternative to resolve this problem, the investment review in the regions where childcare and medical facilities are lacking considers the relevant factors. Furthermore, as mentioned earlier, safety and environment issues have emerged recently, and various supporting projects and policy projects are actively carried out to resolve these issues, which is deemed to be affecting the investment reviews. Therefore, a weight is given with respect to the investment review passage for regions where the safety and environmental areas are relatively vulnerable. Second, in the economic vitalization aspect, it is aimed to prevent decline or falling of the economy in the regions where the number of businesses is declining. To this end, regional vitalization based on joint investment is taken into consideration to create the environment to revitalize the commercial areas and economy and support sustainable operations while suppressing the decline in the number of businesses. Indeed, there are sectors where these two items overlap. However, details on this aspect will be presented in future studies. This is because the names of the specific indicators are different, and because this study seeks to identify the factors that contributed to the passage rate of balanced development indicators within the large scope of project sectors. Furthermore, additional verification is necessary to determine whether the above results are areas that need to be additionally represented in economic analysis. Therefore, this study conducted verification using comprehensive calibrations for each sector.

## 4.2. Verification of Weights by Investment Project Sectors

To verify the above analysis results, investment projects were divided into a total of 14 sectors, and fuzzy AHP techniques were used for analysis (see Figure 1). Because AHP only reflects relative importance, it makes the identification of absolute figures for individual factors difficult. To remedy this, a comprehensive calibration was attempted utilising absolute importance. A structure chart comprising up to four layers was used to derive the weight of each sector. This hierarchical structure chart was developed by referring to the previous studies discussed in Section 2 as well as through focus group and in-depth interviews of various experts. The sample was selected based on expertise in local financial investment projects as well as fields of responsibility or previous research. The survey was administered to a total of 27 university professors (42.85%), 16 researchers (25.39%) from government-run or local government research institutes, 7 chief executive officers or employees (11.11%) of architectural offices or engineering companies having previously participated in feasibility studies, and 13 members of expert groups or government officials (20.63%). The above sample was determined to be representative of the results of the sector-specific weighting of financial projects for the investment review presented in this study.

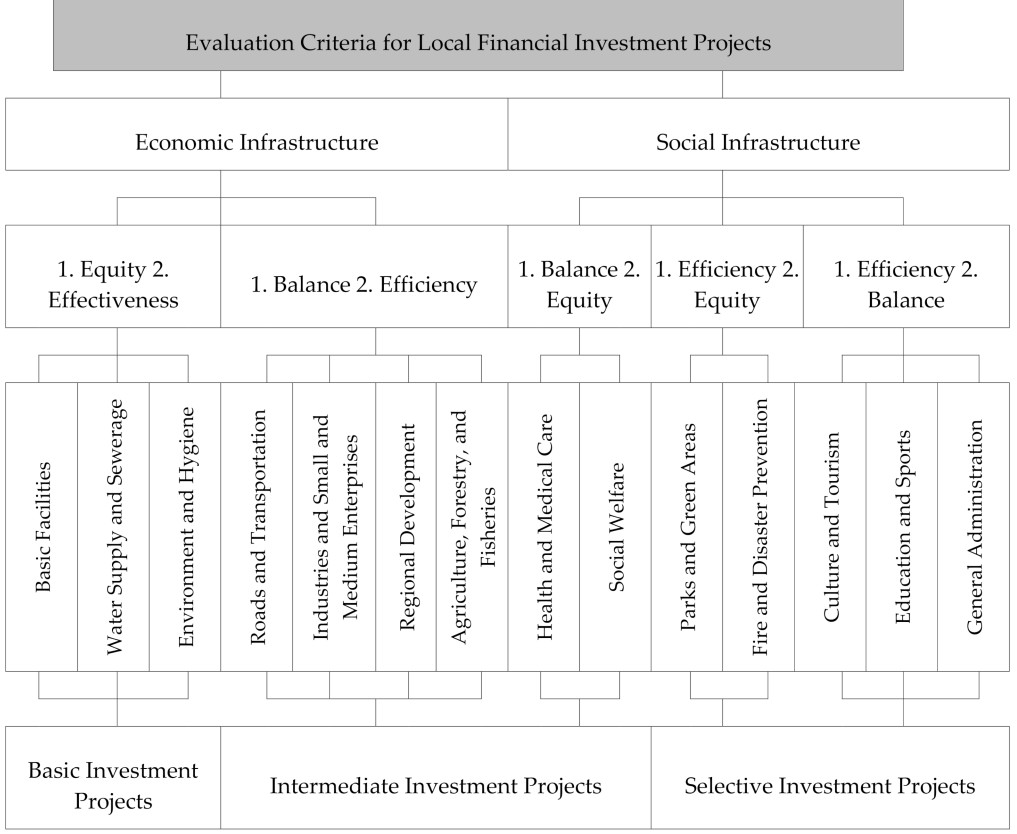

**Figure 1.** Hierarchical Structure Chart. This revised and supplemented hierarchical structure chart is based on the study by Shim [36] on the application of sector-specific weights in the investment review of local financial projects.

The final calibration results based on the elements of local financial investment projects are presented in Table 6. The value of λ on Sugeno's λ-fuzzy scale and the scale factor c are also included in the table. In addition, the rankings of the calibrated values are not absolute rankings in accordance with the size of the values, but they are rankings intended for the efficient utilisation of limited resources.

**Table 6.** Final Calibration Results.

| Class 1 | AHP | Fuzzy Scale | Calibrated Value | Class 2 | AHP | Fuzzy Scale | Calibrated Value | Class 3 | AHP | Fuzzy Scale | Calibrated Value |
|---|---|---|---|---|---|---|---|---|---|---|---|
| Economic infrastructure | 0.415 | 0.815 | 0.658 −0.959 * 1.585 ** | 1. Equity 2. Effectiveness (basic investment projects) | 0.524 | 0.730 | 0.774 −0.877 * 1.478 ** | Basic facilities | 0.302 | 0.757 | 0.517 −0.986 * 1.712 ** |
| | | | | | | | | Water supply and sewerage | 0.325 | 0.802 | 0.556 −0.986 * 1.712 ** |
| | | | | | | | | Environment and hygiene | 0.373 | 0.750 | 0.639 −0.986 * 1.712 ** |
| | | | | 1. Balance 2. Efficiency (intermediate investment projects) | 0.476 | 0.751 | 0.704 −0.877 * 1.478 ** | Roads and transportation | 0.263 | 0.691 | 0.592 −0.979 * 2.251 ** |
| | | | | | | | | Industries, and small and medium enterprises | 0.282 | 0.633 | 0.635 −0.979 * 2.251 ** |
| | | | | | | | | Regional development | 0.239 | 0.626 | 0.538 −0.979 * 2.251 ** |
| | | | | | | | | Agriculture, forestry, and fisheries | 0.216 | 0.581 | 0.486 −0.979 * 2.251 ** |
| Social infrastructure | 0.585 | 0.847 | 0.927 −0.959 * 1.585 ** | 1. Balance 2. Equity (intermediate investment projects) | 0.388 | 0.724 | 0.738 −0.980 * 2.184 ** | Health and medical care | 0.442 | 0.809 | 0.676 −0.918 * 1.530 ** 5 *** |
| | | | | | | | | Social welfare | 0.558 | 0.743 | 0.854 −0.918 * 1.530 ** 1 *** |
| | | | | 1. Efficiency 2. Equity (selective investment projects) | 0.286 | 0.743 | 0.625 −0.980 * 2.184 ** | Parks and green areas | 0.441 | 0.613 | 0.606 −0.804 * 1.374 ** |
| | | | | | | | | Fire and disaster prevention | 0.559 | 0.763 | 0.768 −0.804 * 1.374 ** 3 *** |
| | | | | | | | | Culture and tourism | 0.397 | 0.691 | 0.776 −0.952 * 1.954 ** 2 *** |
| | | | | 1. Efficiency 2. Balance (selective investment projects) | 0.326 | 0.763 | 0.720 −0.980 * 2.184 ** | Education and sports | 0.353 | 0.685 | 0.690 −0.952 * 1.954 ** 4 *** |
| | | | | | | | | General administration | 0.250 | 0.626 | 0.489 −0.952 * 1.954 ** |

In the calibration results, * is the lambda value ($\lambda$), ** is the fuzzy constant (c), and *** is the comprehensive fuzzy analytic hierarchy process (AHP) calibration ranking. Data: Shim, 2020 [36].

The final comprehensive calibration results show that social welfare (0.854), culture and tourism (0.776), and fire and disaster prevention (0.768) were the priorities, and these results are mostly consistent with the analysis results presented in Section 4.1 above. The results reported in the previous subsection show that the sectors of social welfare, safety, and health had positive (+) B values, which is consistent with the priorities suggested by the calibration analysis, indicating the relatively high importance and necessity for representation of these sectors. Therefore, believing that basic infrastructure and minimal safety should be ensured for regional extinction prevention and balanced development, high weights are given to the social welfare and the fire and disaster prevention projects. Meanwhile, the culture and tourism sector is determined to be the most suitable sector to stimulate the local economy and attract tourists or external resources to the region, in conjunction with the current government policy of the social overhead capital provision. This result suggests that it is necessary to consider measures to establish a new economic analysis system for feasibility studies using sector-specific calibration, as entities such as the Board of Audit and Inspection of Korea (2012) have noted various issues regarding unified criteria.

## 5. Conclusions

Compared to preliminary feasibility studies, there was no significant regional deviation in the passage rates of the central investment review for feasibility studies under the Local Finance Act. Therefore, this study sought to identify the factors reviewed in consideration of regional deviation. To this end, the results of the central investment review were utilised to derive measures to improve the local investment review considering balanced regional development. In addition, different analysis methods were used to verify the results. Prior to the full empirical analysis, domestic preliminary feasibility studies in South Korea as well as foreign cases were reviewed for applicability. This is because the representation of economic analysis in terms of actual balanced regional development varies from country to country; hence, it was necessary to verify circumstances specific to South Korea. The results of this verification revealed that most international cases applied various economic analysis methods in limited sectors with regard to balanced regional development. Moreover, in South Korea, concerns have been raised that sector-specific characteristics have not been represented. Therefore, this study sought to verify such issues. Accordingly, balanced development indicators published by NABIS in 2019 were used to identify factors that affected the review—although they were not represented in the actual guidelines or economic analysis—after examining the passage status of projects commissioned for the central investment review in 2019 by region (city, county, and district). As a result, it was confirmed that, among the factors demonstrating significant values, those with positive (+) B values were given greater weight during the investment review. The factors were categorised into the sectors of safety, health, and social welfare. To verify this, comprehensive calibrated values were derived using fuzzy AHP to provide additional sector-specific weights. Consequently, social welfare, fire and disaster prevention, and culture and tourism were found to rank high in the order of priority, thus verifying the above analysis results. Therefore, it can be concluded that additional weights or factors must be considered for areas that have been verified in the two analyses reported here. However, it is necessary to consider from various angles as to how to apply the corresponding factors. This is because the shapes and environments of regions vary from each other and in-depth consideration is required to apply them consistently across the board. For example, the review should be conducted from various perspectives based on this study, such as score and indicator development through degree of regional decline related to balanced regional development, assignment of priorities to regional projects for balanced regional development, extended cost and benefit analysis, and verification and segmentation of regional development rankings by city and province. This study is significant in that it identified balanced development indicators affecting the investment review, which have not been empirically analysed thus far, and verified the indicators by sector. In addition, the findings provide basic data that can be used in future feasibility studies or investment reviews. However, adjustments across sectors, cities, counties, and project sizes were not precisely considered because of

the difficulty of quantitative judgements thereof. Therefore, the limitation of this study is that it does not provide any specific evaluation method or guideline to apply the analysis results in actual practice. Therefore, continuous research must be conducted to steadily improve issues in economic analyses and feasibility studies in the future, and to present concrete research results. In addition, since the forms and environments of each local government differ, preliminary research for the application of such results must be implemented with caution through meticulous analysis and verification by relevant local governments.

**Author Contributions:** H.S. conceived, designed, analyzed, and wrote this paper. J.K. advised on this research, from concept to writing. All authors have read and agreed to the published version of the manuscript.

**Funding:** This study received no external funding.

**Conflicts of Interest:** The authors declare no conflict of interest.

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
