# Peer review of "Measures for the Improvement of Feasibility Studies and Investment Reviews: Identification and Verification of Major Project Sectors Considering Balanced Regional Development"

_sustainability, doi:10.3390/su12229531_

Round 1

Reviewer 1 Report

Authors could consider removing figure 1. I feel that descriptive explanation is enough.

Title 1.1, 1.2 and 2.1 could also be removed.

Please check table 1. Some non English items are there. 

Author Response

I submit the response to the reviewer as an attachment (MS-Word).

Reviewer 2 Report

The subject of the paper is very interesting as feasibility studies and investment Review have always been an important topic on researchers agenda with the aim to provide the best tools for managers and decision makers. When the balanced regional development aspect are inserted in the analysis the decision is even harder and the choice must be supported by good reasons.

From this point of view I believe the article will spark the interest of some readers interested in the subject.

The paper respects the requirements of an academic research article.

The literature review presents most of the relevant aspects on the analyzed topic

The methodology is well presented and suited for the research along with the chosen variables.

The conclusions are supported by the results.

Line 282 The table contains symbols that are not translated in English.

Author Response

(The authors gave the same response as above.)

Reviewer 3 Report

Thank you for the interesting paper. Certainly, it targets a specific audience but given the lack in similar studies it can serve as a solid basis to be used in the evaluation of local/regional investment proposals.

My recommendation to the Editors is to accept the paper subject to minor revisions. My two comments are as follows:

  1. The results as well as the diagnostic statistics derived from the modeling exercise and the econometric analysis are merely mentioned in the text. I would expect that the authors elaborate further on the econometric results and their significance.
  2. In the conclusion section, I think that the authors need to discuss and elaborate on how their results can be utilized in other jurisdictions or geographies in order to benefit the development of regional financial investments.

Author Response

(The authors gave the same response as above.)
